# Genome-wide identification of the MIOX gene family and their expression profile in cotton development and response to abiotic stress

**Zhaoguo Li**[1,2☯], **Zhen Liu**[2☯], **Yangyang Wei**[2], **Yuling Liu**[2], **Linxue Xing**[2], **Mengjie Liu**[2], **Pengtao Li**[2], **Quanwei Lu**[2]*, **Renhai Peng**[1,2]*

**1** Zhengzhou University, Zhengzhou, Henan, China, **2** Research Base, Anyang Institute of Technology, State Key Laboratory of Cotton Biology, Anyang, Henan, China

☯ These authors contributed equally to this work.
* 13707667581@163.com (QL); aydxprh@163.com (RP)

**Data Availability Statement:** All relevant data are within the paper and its Supporting information files.

## Abstract

The enzyme myo-inositol oxygenase (MIOX) catalyzes the myo-inositol into glucuronic acid. In this study, 6 MIOX genes were identified from all of the three diploid cotton species (*Gossypium arboretum*, *Gossypium herbaceum* and *Gossypium raimondii*) and *Gossypioides kirkii*, 12 MIOX genes were identified from two domesticated tetraploid cottons *Gossypium hirsutum*, *Gossypium barbadense*, and 11 MIOX genes were identified from three wild tetraploid cottons *Gossypium tomentosum*, *Gossypium mustelinum* and *Gossypium darwinii*. The number of MIOX genes in tetraploid cotton genome is roughly twice that of diploid cotton genome. Members of MIOX family were classified into six groups based on the phylogenetic analysis. Integrated analysis of collinearity events and chromosome locations suggested that both whole genome duplication and segmental duplication events contributed to the expansion of MIOX genes during cotton evolution. The ratios of non-synonymous (Ka) and synonymous (Ks) substitution rates revealed that purifying selection was the main force driving the evolution of MIOX genes. Numerous cis-acting elements related to light responsive element, defense and stress responsive element were identified in the promoter of the MIOX genes. Expression analyses of MIOX genes based on RNA-seq data and quantitative real time PCR showed that MIOX genes within the same group shared similar expression patterns with each other. All of these results provide the foundation for further study of the biological functions of MIOX genes in cotton environmental adaptability.

## Background

Myo-inositol (MI) is a small molecule that is important in many different developmental and physiological processes in eukaryotic cells [1]. MI participates in the production of stress-related molecules, cell wall biosynthesis, phytic acid biosynthesis, auxin transport and storage [2]. The Myo-inositol oxidation pathway (MIOP) effectively control MI homeostasis. In this pathway, Myo-inositol oxygenase (MIOX, E.C. 1.13.99.1) is a key enzyme that catalyzes the conversion of MI into D-glucuronic acid (D-GlcUA), which is subsequently activated to

**Funding:** This work is partially support by College Student Innovation and Entrepreneurship Training Program in Henan Province(S202011330003), China Postdoctoral Science Foundation (2018M632761), the National Natural Science Foundation of China (No. 31471548), the National Key Research and Development Program of China (No. 2018YFD0100302) and the Science and Technology Development Project of Anyang City (2018-66-117). The funders had no role in the design of the study and collection, analysis, and interpretation of data and in writing the manuscript.

**Competing interests:** The authors have declared that no competing interests exist.

UDP-D-glucuronic acid (UDP-GlcA) and serves as an important precursor for plant cell wall polysaccharides [3, 4]. Furthermore, previous reports suggested that MIOX might also be involved in the production of ascorbate, consequently, protection from ROS-mediated injury [5]. Studies have shown that MIOX plays a role in the response to environmental stresses. In *Arabidopsis*, MIOX was encoded by a gene family of four members, it has been reported that *AtMIOX4* over expression can increase the tolerance to cold, heat and salt [3, 6–8]. In addition, it has also been reported that rice MIOX has a specific function in drought stress tolerance by decreasing oxidative damage [9].

Cotton (*Gossypium*) is one of the most economically important fiber crop plants throughout the world. It is a remarkably diverse genus, with over 50 species divided into eight diploid genomic groups, designated A-G and K, and one tetraploid genomic group, namely AD [10]. Divergence analysis suggests that the major diploid branches of the cotton genus diverged about 7~11 million years ago, and the polyploid clade originated circa 1~2 million years ago [11–14]. Due to the rapid development of next generation sequencing, the whole genomes have been reported for the diploid D-genome of *G. raimondii* [15], A-genome of *G. arboretum* [16], *G. herbaceum* [17], and allopolyploid AD genome species of *G. hirsutum* [18], *G. barbadense* [18], *G. darwinii*, *G. mustelinum* and *G. tomentosum* [19]. The sequence of these genomes provides useful resources for studying the gene families and a series of families have been genome-wide analyzed in cotton [20, 21].

Although cotton has better abiotic stress tolerance compared with other crops, its yield and fiber quality under saline-alkali, drought and heat stress are still under great threat. Considering the role of MIOX gene in various abiotic stress responses in plants and there is no genome-wide identification and characterization of MIOX gene family members in cotton. We conducted a systematic analysis of the cotton MIOX gene family by a genome-wide search, including the phylogeny evolutionary relationships, group classification, conserved motifs and the expression profile etc. Our results will provide good foundation for understanding the key roles of MIOX genes in cotton stress-responsive and other biological processes.

## Results

### Identification of the MIOX gene family

A total of 81 MIOX proteins were identified from nine species, including 75 MIOX proteins in eight *Gossypium* genomes and 6 MIOX proteins in the *Gossypioides kirkii* genome. Except for one MIOX protein from *G. herbaceum* has significantly more amino acids (896 aa) and two MIOX proteins from *G. barbadense* has significantly less amino acids (101 aa and 113 aa), the others are relatively conservative (S1 Table). Furthermore, we found that all of the three diploid *Gossypium* species and the *Gossypioides kirkii* contained 6 MIOX proteins, the number of MIOX proteins in tetraploid cotton was roughly twice that of diploid cotton, but the three wild tetraploid cottons (*G. tomentosum*, *G. mustelinum* and *G. darwinii*) missed a MIOX protein.

For convenience, we renamed the MIOX gene family. Gh, Gb, Ga, Gher, Gr, Gd, Gt, Gm, and Gkk were used as prefixes before the names of MIOX genes from *G. hirsutum*, *G. barbadense*, *G. arboreum*, *G. herbaceum*, *G. raimondii*, *G. darwinii*, *G. tomentosum*, *G. mustelinum* and *Gossypioides kirkii* respectively. We assigned the names to MIOX proteins as (*GhMIOX01-GhMIOX12*) for *G. hirsutum*, (*GbMIOX01—GbMIOX12*) for *G. barbadense* and so on. The new names of the sequence are only used in this study.

### Phylogenetic analysis of the MIOX gene family

To reveal the evolutionary relationship of identified MIOX proteins, the amino acid sequences of 75 proteins from *Gossypium* species, 6 proteins from *Gossypioides kirkii* and 4 proteins from

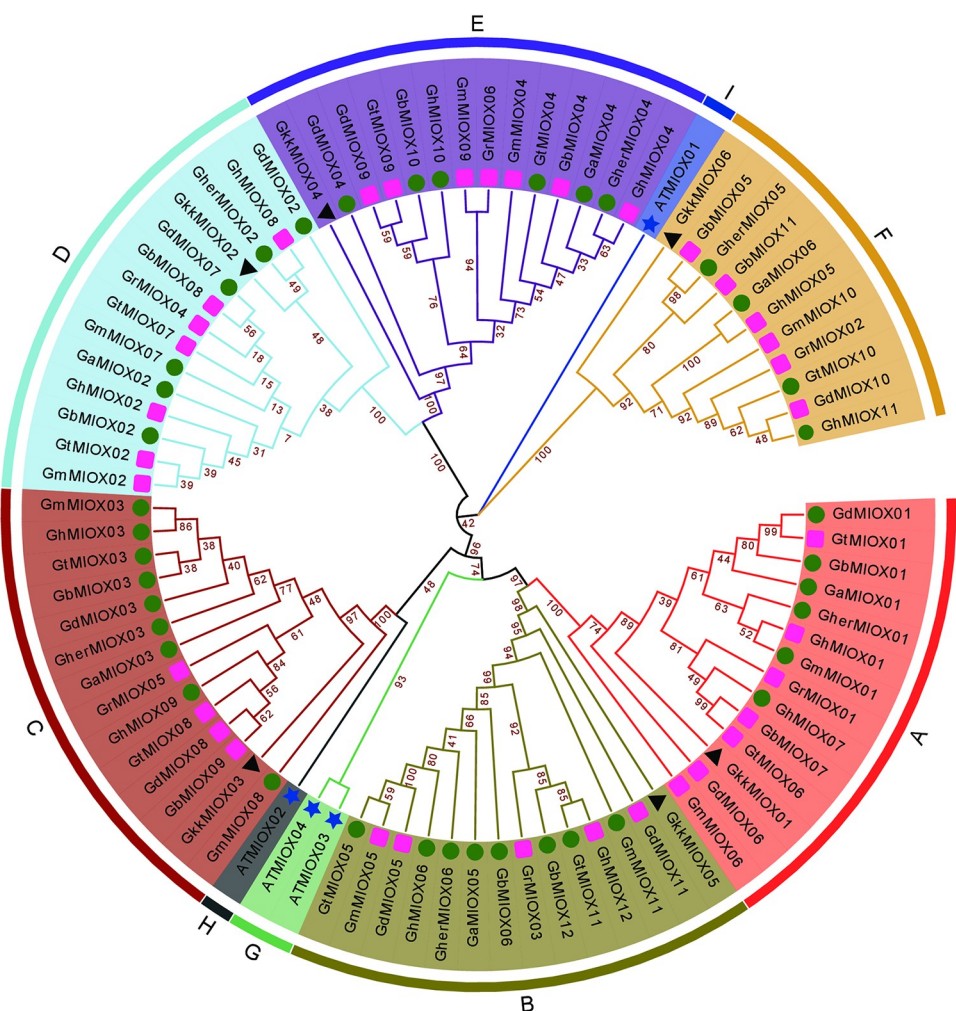

**Fig 1. Phylogenetic tree of MIOX proteins from 8 *Gossypium* species, *Gossypioides kirkii* and *Arabidopsis thaliana*.** The different-colored arcs indicate different groups of MIOX proteins. The green circles and red boxes represent MIOX proteins from A genome and D genome, respectively, black triangles represent MIOX proteins from *Gossypioides kirkii*, blue stars represent MIOX proteins from *Arabidopsis thaliana*.

*Arabidopsis thaliana* were used to construct a phylogenetic tree (Fig 1, S1 Fig). According to the result, 81 MIOX proteins from 8 *Gossypium* species and *Gossypioides kirkii* were divided into six groups of A-F, and the number of MIOX proteins was very stable in each group, diploid species containing 6 MIOX proteins had one distribution in each group, while tetraploid cottons containing 11 or 12 MIOX proteins had one or two distribution in each group. All of these genes showed one-to-one homology relationship among different genomes or subgenomes. Compared to the A~F groups, the group G~I only contained *A. thaliana* MIOX proteins.

## Gene structure and protein motifs of the MIOX gene family

To explore the structural diversity of MIOX genes, the intron-exon organization of each MIOX gene was analyzed (Fig 2). The number of introns ranged from 3 to 21, most (53/81) MIOX genes contained 9 or 10 introns. In the same group, the intron numbers exhibited comparable exon-intron structure and intron numbers, while many *Gossypioides kirkii* MIOX

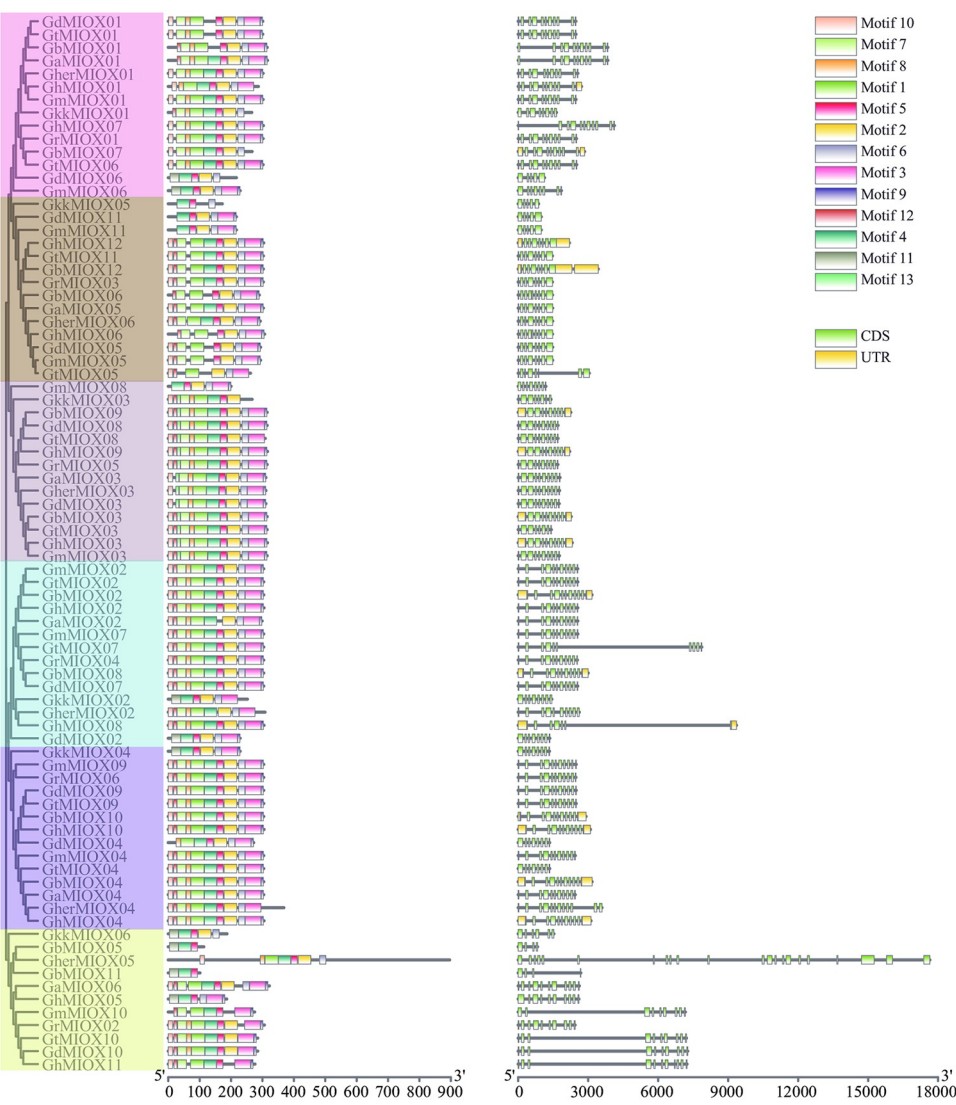

**Fig 2. The exon–intron structures and conserved motifs of MIOX.** The length of proteins and DNA sequence was estimated using the scale at the bottom, black line indicated the non-conserved amino acid or introns. (a) The conserved motifs of MIOX gene family; (b) The gene structure of MIOX gene family.

genes were different from cotton, implying important evidence for the phylogenetic relationship among members of the MIOX gene family.

Furthermore, we investigated the conserved motifs in MIOX proteins to understand the diversity of motif compositions (Fig 2). A total of 13 motifs, named Motif 1~Motif 13, were identified in MIOX proteins. The number of motifs varied from 3 to 12 in each MIOX protein and most MIOX proteins within the same group exhibited similar motif composition and arrangement, which indicates that the members of MIOX gene family that clustered in the same group may have similar biological functions. Motif 5 was found in 78 MIOX proteins, Motif 8 was completely missing only in group B, and Motif 13 was specific to group C. The gene structure and motif composition of the MIOX members from each group that obtained from phylogenetic analysis were similar, which indicates the classification was reliable.

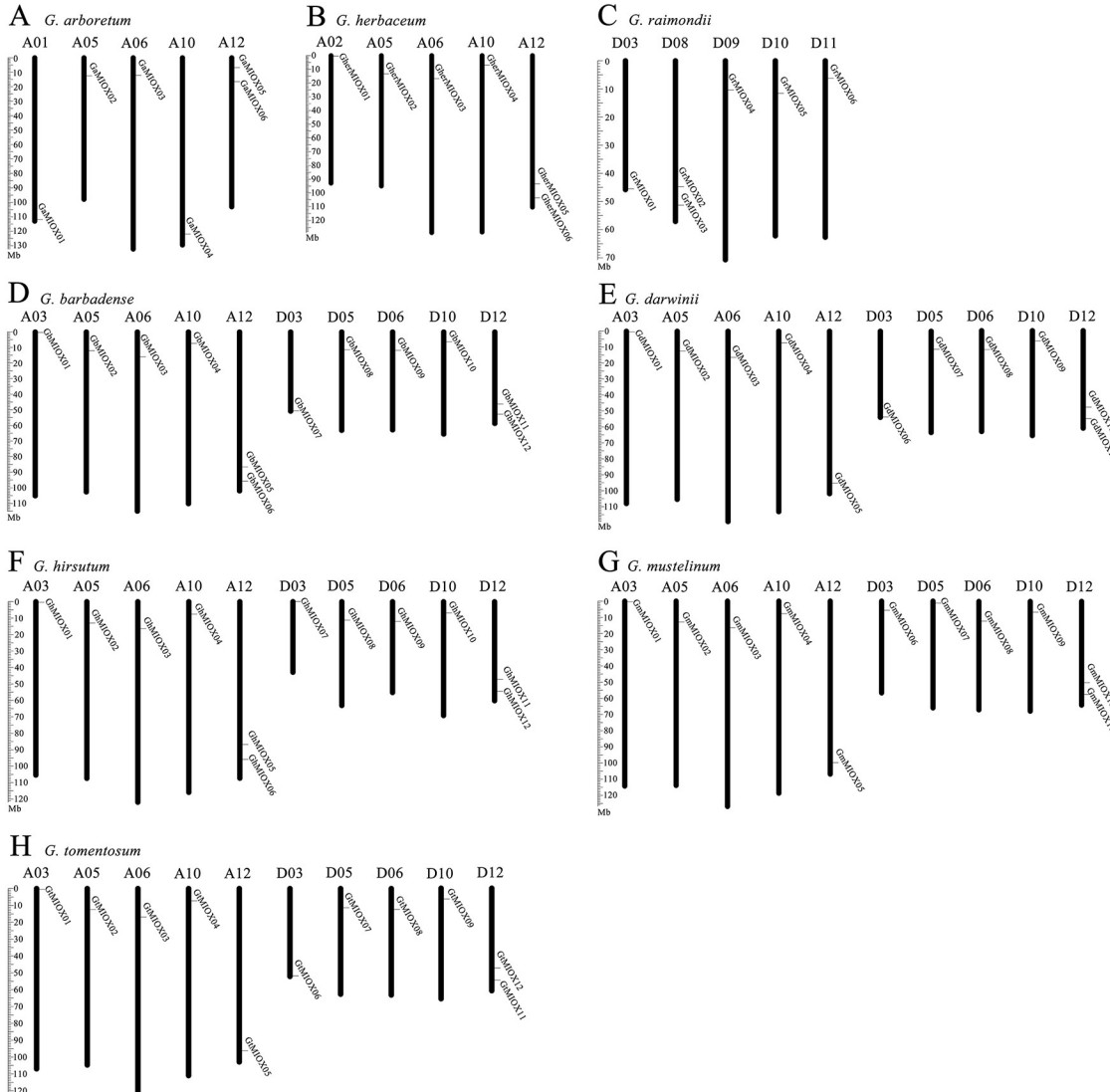

**Fig 3. Distribution of the MIOX genes on the cotton chromosomes.** The vertical bar located on the left side indicates the chromosome sizes in mega bases (Mb), the chromosome number is located above each chromosome.

## Chromosomal location and gene duplications of the MIOX gene family

To more intuitively understand the distribution of MIOX family genes on the chromosomes, we performed a chromosomal localization analysis. The result shows that MIOX genes were mapped onto 5/10 different chromosomes of diploid/tetraploid cotton. Each chromosome contained only one MIOX gene usually, while some chromosome contained 2 MIOX genes (Fig 3). In addition, the distribution of *Gossypioides kirkii* MIOX genes showed similar to *Gossypium*.

Whole genome duplication, segmental duplication and tandem duplication provides major forces that drive the expansion of gene families. The number of MIOX genes in tetraploid *Gossypium* species was twice as much as that in diploid *Gossypium* species which indicates the expansion of MIOX gene family during polyploidization. We searched the segmental duplication using MCScanX within each genome, and identified 143 collinear genes pairs among the

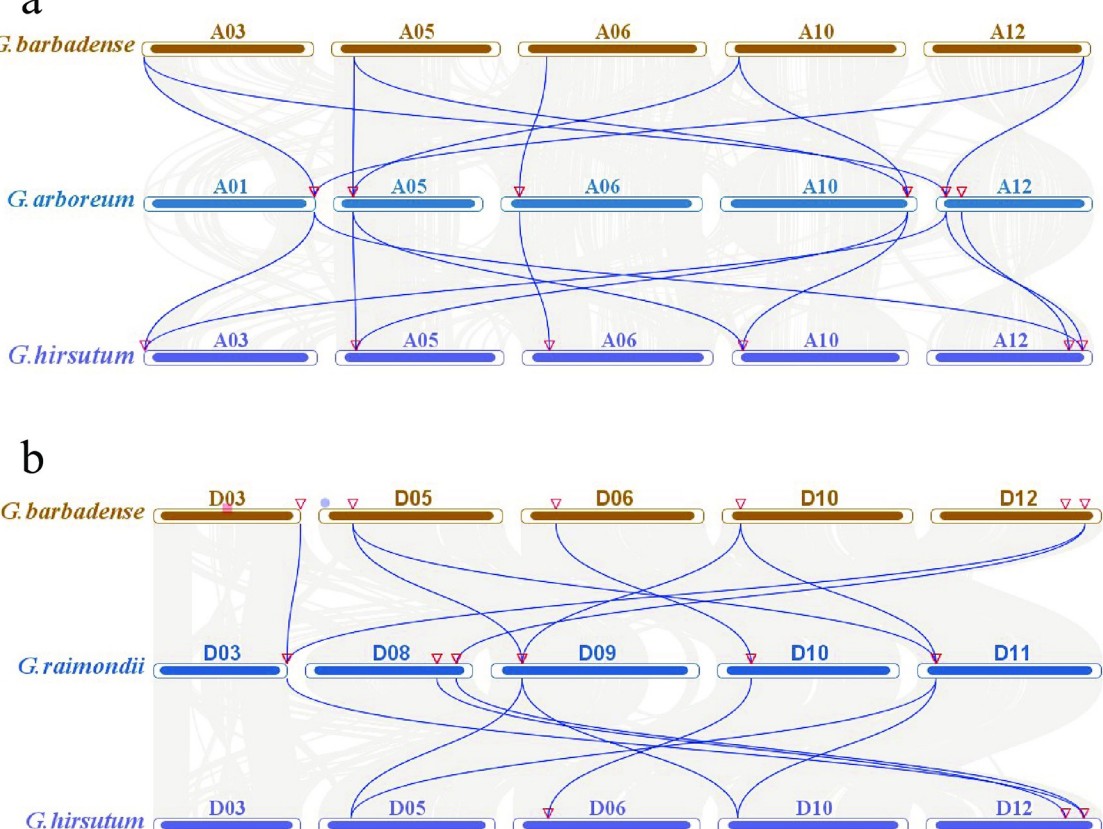

**Fig 4. Collinearity of MIOX genes between different cotton species.** (a) Collinearity between A-genome of *G. arboretum* and A-subgenome of *G. hirsutum* and *G. barbadense*; (b) Collinearity between D-genome of *G. raimondii* and D-subgenome of *G. hirsutum* and *G. barbadense*.

72 MIOX genes. Of the 75 MIOX genes in eight *Gossypium* genomes, only 3 were located outside of the duplicated blocks, while 96% (72 of 75) were located in duplicated regions. In addition, 60, 38 duplication gene-pairs were found between diploid *G. arboretum* A-genome and tetraploid *G. hirsutum*, *G. barbadense* A-subgenome respectively. 50, 17 duplication gene-pairs were found between diploid *G. raimondii* D-genome and tetraploid *G. hirsutum*, *G. barbadense* D-subgenome respectively (Fig 4).

According to Holub's description, a chromosomal region within 200 kb containing two or more genes was defined as a tandem duplication event [22]. Our results indicated that the MIOX genes of the nine species have no tandem duplication. Furthermore, the role of transposable elements also play an important role in the expansion of many genes in many plants [23–25]. However, our results show that none of MIOX gene was amplified by transposable elements.

## Selection pressure of the MIOX gene family

To further elucidate the selection pressure of functional diversification of each MIOX group, we have comprehensively evaluated the non-synonymous (Ka) and synonymous (Ks) substitution rates (Fig 5). Generally, the value of Ka/Ks represents the ratio between Ka and Ks of two homologous protein-coding genes. Ka/Ks > 1 indicates that a gene has been positively selected, while a Ka/Ks = 1 indicates that a gene has been neutrally selected, and a Ka/Ks < 1

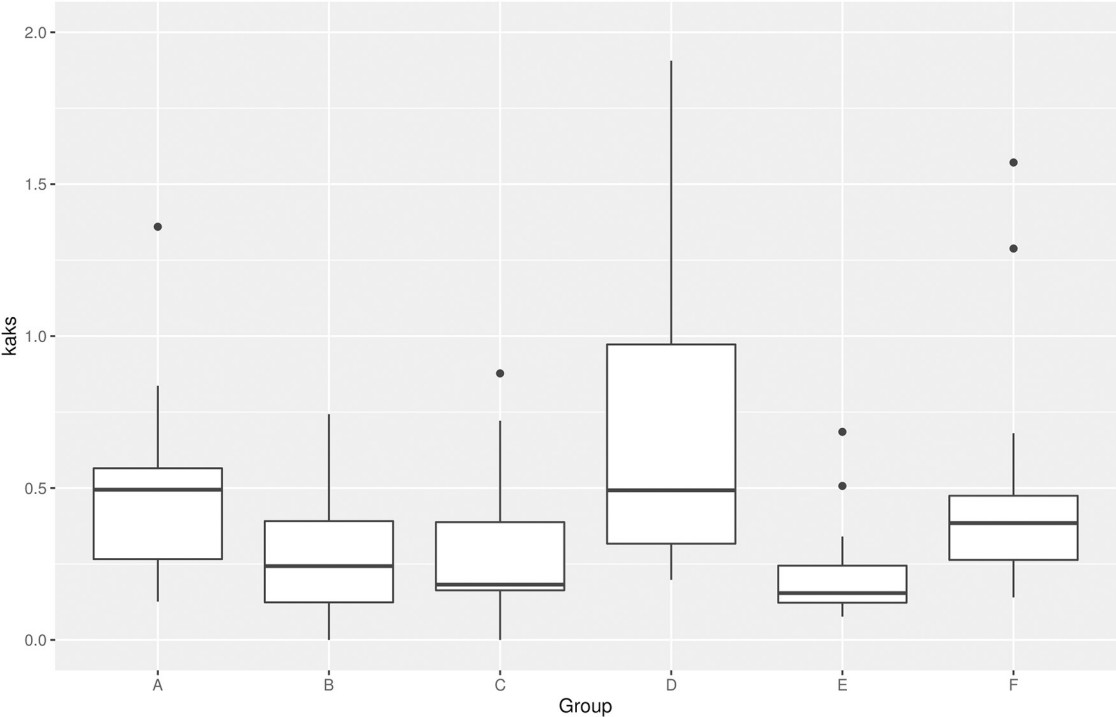

**Fig 5. The Ka/Ks ratio of MIOX orthologous gene pairs of each group.**

indicates that a gene has been selectively purified. Our results showed that the median Ka/Ks ratios of the 6 groups ranged from 0.164324 (Group E) to 0.500071 (Group A), and except for very few Ka/Ks ratios >1, majority of Ka/Ks ratios were <1. These results indicated that MIOX gene family was under purifying selection and might play specialized roles in the adaptive evolution of cotton.

## Cis-elements in the promoter of MIOX genes

Cis-elements in promoters play vital roles in regulating the expression of genes. To gain more insight into the functions of MIOX genes, the cis-regulatory elements were scanned in the 2000 bp upstream of the transcription start sites of cotton MIOX genes (Fig 6, S2 Table, S2 Fig). The results showed that there were many kinds of response elements, such as light responsive element, defense and stress responsive element involved in drought and low temperature, and hormone responsive element associated with salicylic acid, abscisic acid, gibberellin and MeJA. All of the cotton MIOX genes contained more than one light responsive element, however only 29.33% (22/75) of the cotton MIOX genes contained auxins responsive element.

The cis-elements of MIOX genes in the same phylogenetic group were similar. In addition, within the same group, half of the cis-elements of MIOX genes in tetraploid *Gossypium* species are similar to the diploid A genome species (*G. herbaceum* and *G. arboretum*), and half are similar to the D genome species (*G. raimondii*). These results further indicated the expansion of MIOX gene family during polyploidization.

## Tissue-specific expression profiles of MIOX genes

To study the tissue-specific expression patterns of the MIOX genes, we analyzed the expression profiles of the MIOX genes in different tissues. As shown in Fig 7, MIOX genes showed

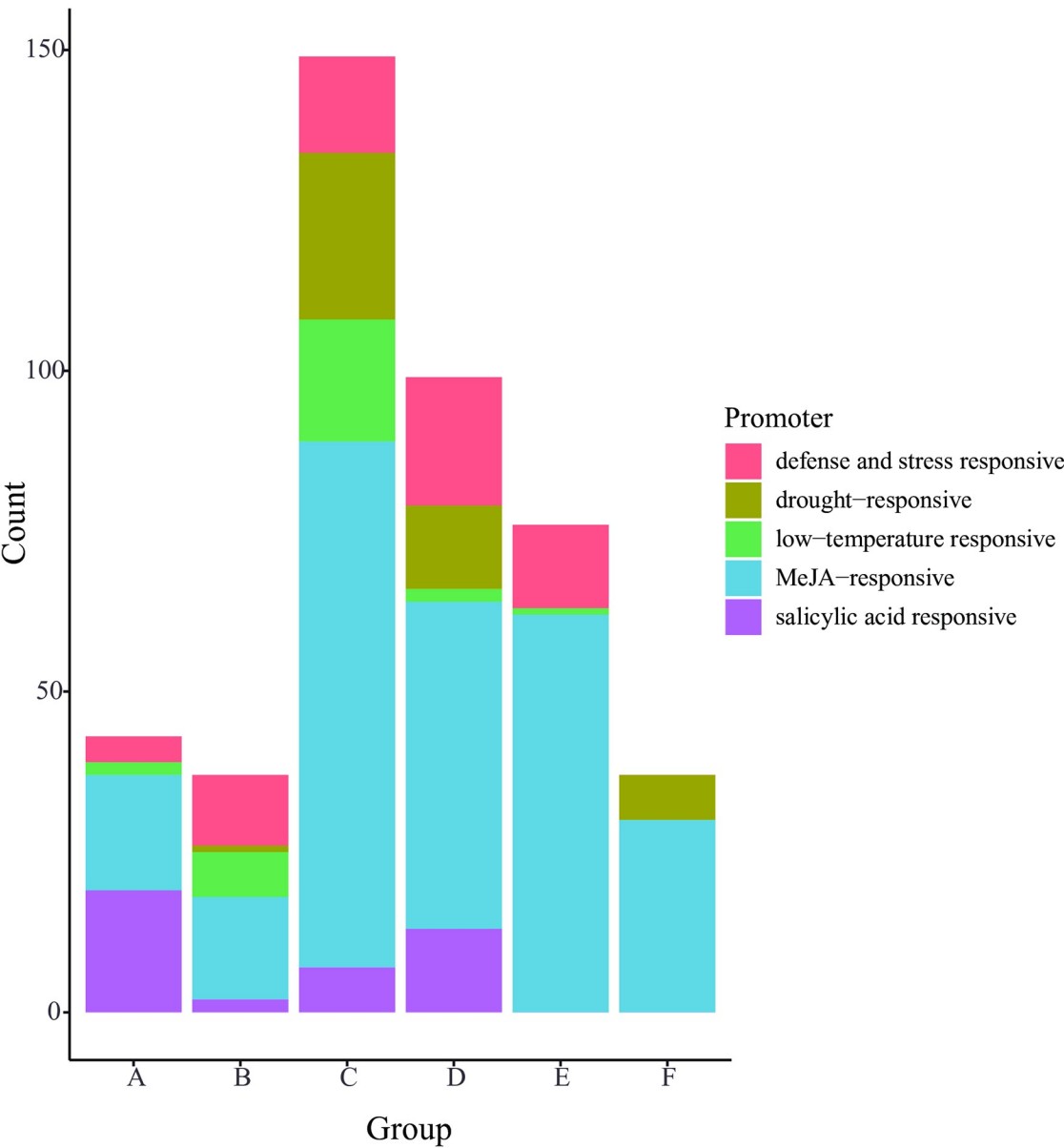

**Fig 6. Number of MIOX gene cis-elements in each group.**

different expression levels in different tissues. In *G. hirsutum*, the expression of *GhMIOX02* and *GhMIOX08* were higher in leaf, torus, pistil, bract and sepal, while their expression was lower in root. The expression of *GhMIOX04* and *GhMIOX10* was higher in root, leaf, torus and sepal, while their expression was lower in pistil and bract. In *G. barbadense*, *GbMIOX02*, *GbMIOX08*, *GbMIOX04* and *GbMIOX10* had similar expression profiles with *G. hirsutum*. In addition, *GhMIOX03* of *G. hirsutum* showed higher transcription level only in root and stem, however, *GbMIOX03* of *G. barbadense* showed higher transcription levels in root, stem and pistil.

The expression levels of MIOX genes at ovule and fiber developmental stages were also investigated (Fig 7). At most stages of development, the expression of *GhMIOX08* from *G. hirsutum*, *GbMIOX08* from *G. barbadense* were higher than that of the other genes; Several genes

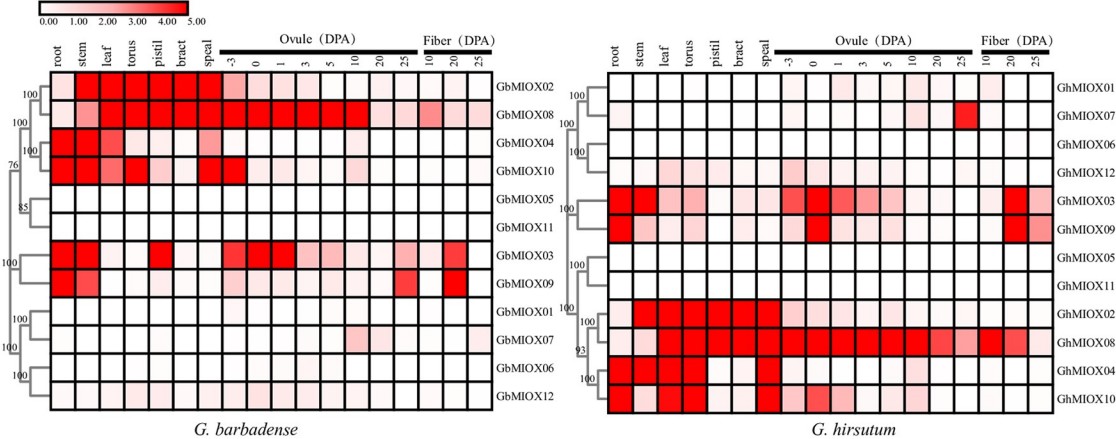

**Fig 7. Expression profiles of MIOX genes in different tissues and during different developmental stages of ovule and fiber.**

were expressed at high levels during specific developmental stages; for instance, *GhMIOX03*, *GhMIOX09*, *GhMIOX10* from *G. hirsutum* and *GbMIOX03*, *GbMIOX09*, *GbMIOX10* from *G. barbadense*. In contrast, the RNA transcript levels of *GhMIOX05*, *GhMIOX06*, *GhMIOX11* from *G. hirsutum* and *GbMIOX01*, *GbMIOX05*, *GbMIOX06*, *GbMIOX11* from *G. barbadense* were low at all stages and all tissues. These findings indicated the MIOX genes play differential roles in tissue development.

## Stress-induced expression patterns of MIOX genes

The expression patterns of MIOX genes were further analyzed in *G. hirsutum* and *G. barbadense* exposed to different durations of cold, heat, salt, and drought stresses for different times by RNA-seq data downloaded from the public database (Fig 8). Based on the clustering analysis, MIOX genes in the same group had similar expression patterns. *GhMIOX02*, *GhMIOX08*,

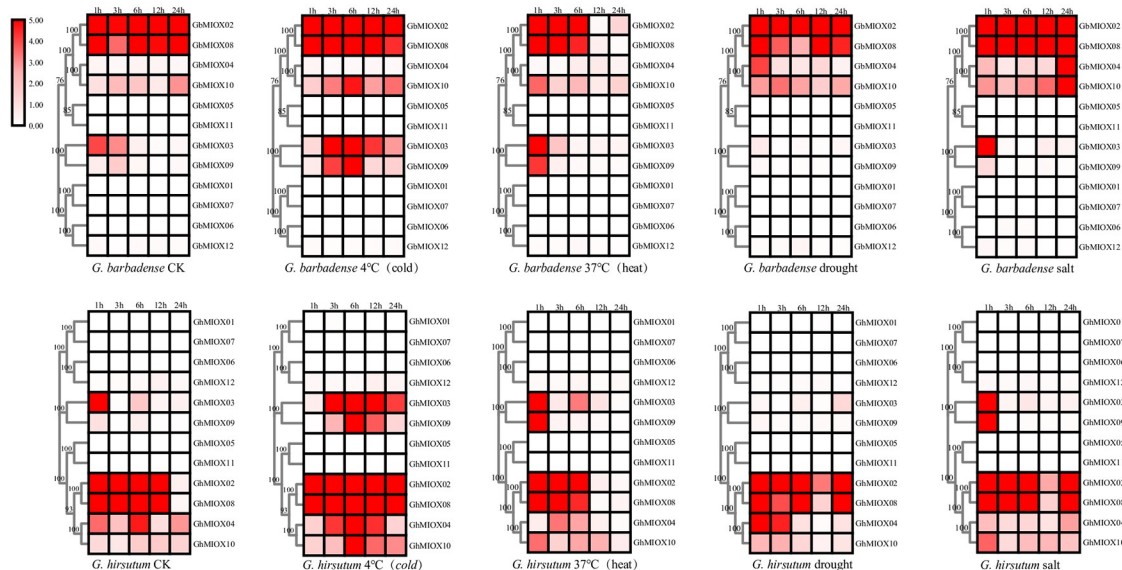

**Fig 8. Expression profiles of the MIOX genes in response to different stresses.**

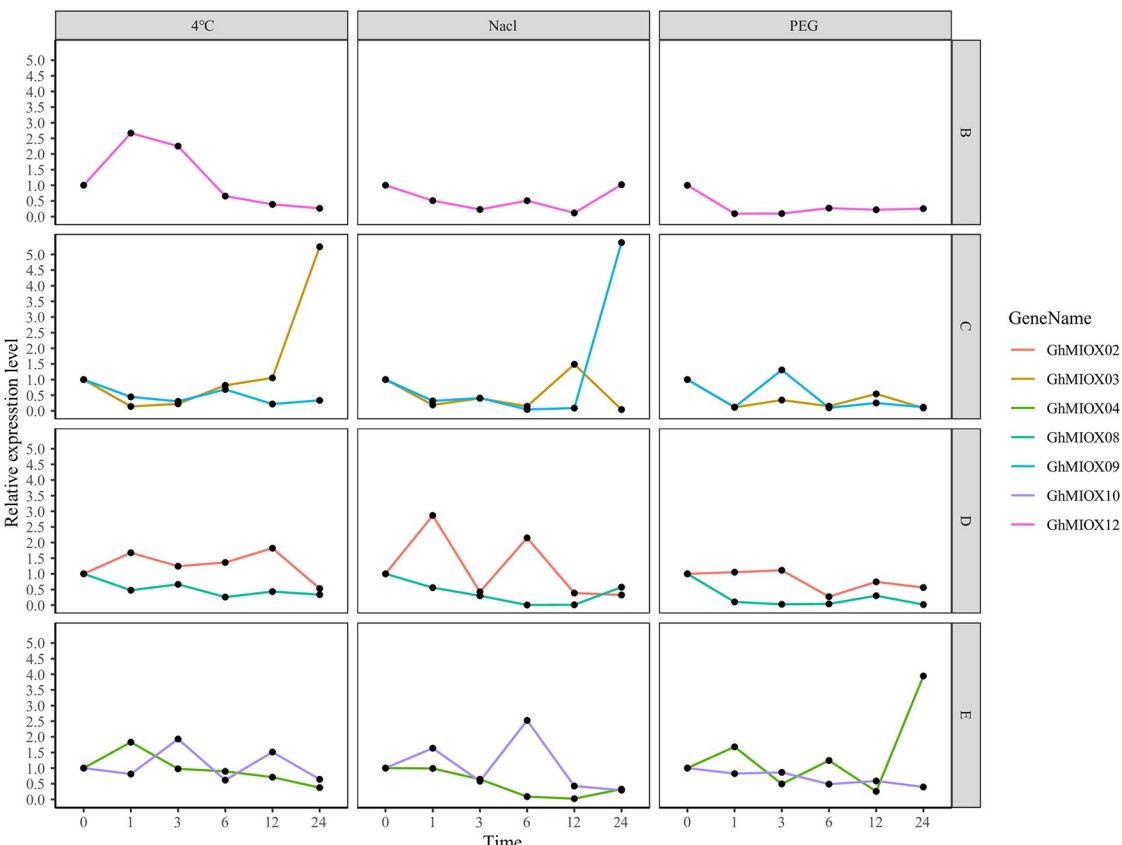

**Fig 9. Expression of MIOX genes under different stress conditions.**

*GbMIOX02* and *GbMIOX08* in group D showed higher expression levels in all of these stresses, while *GhMIOX04*, *GhMIOX10*, *GbMIOX04* and *GbMIOX10* in Group E and *GhMIOX03*, *GhMIOX09*, *GbMIOX03* and *GbMIOX09* in Group C had different expression patterns under different conditions. In addition, the genes in other groups were not significantly responded to these stresses. These findings suggested the functional divergence of the MIOX genes in abiotic stress.

## Expression analysis of MIOX genes under cold, PEG, and NaCl treatments

We analyzed the expression patterns of the *G. hirsutum* MIOX genes after 0, 1, 3, 6, 12, and 24h of cold (4°C), PEG, and NaCl treatment, respectively. The results showed that most of the MIOX genes responded to the different stress treatments (Fig 9). The upregulation (increased 3.9 ~ 5.3 times) of *GhMIOX03*, *GhMIOX09* and *GhMIOX04* reached their peaks after 24h treatment of Cold, NaCl and PEG respectively. The expression levels of *GhMIOX03*, *GhMIOX08* and *GhMIOX12* decreased after 1h of the PEG treatment. *GhMIOX02*, *GhMIOX12* under NaCl treatment and *GhMIOX09* under Cold, PEG conditions showed a down-up-down-regulation trend, and the fluctuating trend between the different genes was significantly different. In addition, some MIOX genes only showed slightly change, for example, *GhMIOX02* response to cold and *GhMIOX10* response to PEG. Therefore, the *G. hirsutum* MIOX genes are widely involved in abiotic stress responses.

## Discussion

MI is a crucial substance in various aspects of plant physiology. Plants maintain an MI pool at a basal level throughout their life cycle, and MIOX is used to control the metabolite level of MI in plants [1, 2, 5, 8]. MIOX proteins are conserved and present in nearly all eukaryotes. MIOX is crucial in abiotic stress tolerance and previous studies on the MIOX gene family have been performed in many plants [2, 5], including *Arabidopsis thaliana* [3, 4], *Oryza sativa* [9] and *Solanum lycopersicum* [26]. In this study, we performed a comprehensive identification of MIOX genes in 5 tetraploid cotton species, 3 diploid cotton species and a cotton closely related species *Gossypioides kirkii*, with an aim of understanding the important and diverse roles of this gene family in abiotic stress tolerance in plants.

The number of MIOX genes in many species was small, for example, 4 MIOX genes were identified from *Arabidopsis thaliana*, 1 MIOX gene from *Oryza sativa* and 5 MIOX genes from *Solanum lycopersicum*. In addition, the number of MIOX genes in different species of *Gossypium* is also small. Our results indicated that no MIOX gene was amplified by tandem duplication and transposable elements, which suggested that the number of MIOX genes is relatively stable in the process of evolution. The number of MIOX genes roughly doubled during *Gossypium* polyploidization, our results further indicated that the amplified pair belong to the same group and maintained similar gene structure, motif composition and expression profile.

An analysis of gene expression patterns can be used to some extent to predict the molecular functions of genes involved in different processes. Our heatmap data showed that most MIOX genes within each group shared similar expression patterns. The MIOX genes in Group D in *G. hirsutum* (*GhMIOX02* and *GhMIOX08*) and *G. barbadense* (*GbMIOX02*, *GbMIOX08*) were expressed in leaf, torus, pistil, bract, speal and responded to cold, heat, salt, and drought stresses. In contrast, the expression of MIOX genes in Group F in *G. hirsutum* (*GhMIOX05*, *GhMIOX11*) and *G. barbadense* (*GbMIOX05*, *GbMIOX11*) were barely observable.

## Conclusions

We performed a genome-wide analysis of the MIOX gene family in 8 cotton species. The MIOX family was divided into 6 groups based on the phylogenetic tree. To understand the expansion mechanism of MIOX family, the gene duplication events was investigated, and the result indicated that segmental duplication and whole genome duplication were the major driving forces of MIOX family expansion, no tandem duplication and transposable element amplification was found. The expression profiles of *G. hirsutum* and *G. barbadense* MIOX genes in different tissues and under abiotic stresses were also analyzed. The results indicated that MIOX genes within the same group shared similar expression pattern and they exhibited different expression levels in different groups. Furthermore, the Ka/Ks ratio suggesting that all groups experienced purifying selection pressure. Our results will provide clues for researchers regarding the evolution and biological functions of MIOXs.

## Methods

### Identification of MIOX genes in cotton

The genome sequences of *G. darwinii* (AD5 HGS_v1.1), *G. mustelinum* (AD4 JGI_v1.1) and *G. tomentosum* (AD3 HGS_v1.1) [19] were downloaded from NCBI (https://www.ncbi.nlm.nih.gov/genome/), the genome sequences of *G. hirsutum* (TM-1 HAU_v1.1) [18], *G. barbadense* (HAU_v2_a1) [18], *G. arboretum*(A2 CRI-updated_v1) [16], *G. herbaceum*(A1 WHU_v1) [17], *G. raimondii*(D5 NSF_v1) [15] and a cotton closely related species

*Gossypioides kirkii* (ISU_v3_a3) [27] were downloaded from COTTONGEN (http://www.cottongen.org). *Arabidopsis* MIOX proteins were used to search possible cotton MIOX sequences by BLASTP with an E-value≤1.0E$^{-3}$ [28, 29]. The HMMER [30] program was used to identify MIOX proteins with the Hidden Markov Model of the MIOX domain (PF05153), which was downloaded from the Pfam database (http://pfam.xfam.org). Furthermore, NCBI CD-Search (https://www.ncbi.nlm.nih.gov/cdd/) and Search Pfam tools (http://pfam.xfam.org/search) were used to confirm the candidate sequences. The number of amino acids, molecular weights and theoretical isoelectric points of MIOX proteins were calculated using the ExPASy online server tool (https://www.expasy.org/).

## Phylogenetic, gene structure, conserved motif and cis-elements analysis of MIOXs

The ClustalX was used to align MIOX protein sequences. A neighbor-joining tree of MIOX proteins was constructed using MEGA-X with 1000 bootstrap replications [31]. The phylogenetic tree was drawn using EvolView [32]. The structure of MIOX genes were analyzed with TBtools [33]. The conserved motifs of MIOX protein sequences were obtained using MEME [34]. To investigate putative cis–acting regulatory elements of MIOX genes, 2000 base pair genomic DNA sequences upstream of the initiation codon were retrieved and screened against the PlantCARE database [35].

## Chromosomal distribution and gene duplication of MIOX genes

The chromosome distributions of MIOX genes were extracted from the GFF files and the Mapchart [36] software was used to visually map the chromosomal location. We made use of Multiple collinear scanning toolkits (MCScanX) [37] to detect the gene duplication events and tandem duplications were identified as previously described. The synonymous (Ks) and non-synonymous (Ka) substitution rates of MIOX genes were calculated by KaKs_Calculator [38].

## Transposable elements expansion analysis of MIOX genes

To investigate whether transposable elements played roles in expansion of the MIOX genes, LTR_retriever [39], LTRharvest [40], LTR_FINDER [41], RepeatModeler and RepeatMasker [42] were used to identify transposable elements. We then compared those results with the genome annotation to predict genes inside transposable elements in the 10 genomes.

## Expression patterns of MIOX genes

To study the expression patterns of MIOX genes, *G. hirsutum* and *G. barbadense* high-throughput transcriptome sequencing data including various tissues, developmental stages and stress treatments were downloaded from the NCBI SRA (PRJNA490626). Trimmomatic [43] was used to remove the adapters and to perform quality control. The program hisat2 [44] was used to map the reads to the genomes, then Fragments Per Kilobase of transcript per Million fragments (FPKM) values of the MIOX genes were calculated by Cufflinks [45, 46]. Heat maps of gene expression profiles were drawn by TBtools [33].

## RNA isolation and real-time PCR analysis of MIOX genes

Total RNA was extracted by using an EASYspin Plus Plant RNA Rapid Extraction Kit (Aidlab Biotech) and RNA integrity and concentration are measured using NanoDrop2000. The first cDNA strand was synthesized with 1 ng total RNA by using a TransScript II All-in-One First-Strand cDNA Synthesis SuperMix for qPCR (TransGen Biotech) based on the manufacturer's

**Table 1. The specific primers for qRT-PCR of _G. hirsutum_ MIOX genes.**

| Gene name | Forward Primer sequences (5'-3') | Reverse Primer sequences (5'-3') |
|---|---|---|
| GhMIOX02 | CTTGCTGCAAACAGCTGAGG | GCTCCCCAAAACCAGGATGA |
| GhMIOX03 | CGAGAAGCCTGAGCTAGTGT | CCACGCTCTTTTGCCTTTCA |
| GhMIOX04 | TCCTCATTGATCAACCTGATTTTGG | CACCCTGTTGCCTCTCACTT |
| GhMIOX08 | CCTCAGTGGGCTGTTGTAGG | TCTCTGTTCTCCAACAGAAGAAA |
| GhMIOX09 | CCTTCAGCAGGGCTGTTCAT | GCCACCTTGCTCTTGCTGTA |
| GhMIOX10 | CTTGCTGCAAACAGCTGAGG | CCTACAACAGCCCACTGAGG |
| GhMIOX12 | CCCATTGCACAAGCATGGAG | CGTGAGCTTTGCTCTTGCTG |

instructions. The qRT-PCR reactions were performed on the ABI 7500 Fast (Applied Biosystems, Foster City, CA, USA) using TransStart Top Green qPCR SuperMix (TransGen Biotech) with specific primers designed by NCBI-Primer-BLAST (Table 1). Three technical replicates were used for each sample. The relative gene expression was calculated with the $2^{-\Delta\Delta Ct}$ method [47].

## Supporting information

**S1 Fig. Alignment of 85 MIOX protein sequence.** Multiple sequence alignments were conducted using ClustalW.
(PDF)

**S2 Fig. Promoter analyses of MIOX genes.** The promoter sequences (2 kb upstream of ATG) of the MIOX genes were analysed by PlantCARE.
(PDF)

**S1 Table. List of the identified MIOX genes in cotton.**
(XLSX)

**S2 Table. Number of cis-elements present in the promoter regions of the MIOX genes.**
(XLSX)

## Acknowledgments

We thank all colleagues in our laboratory for helpful discussions and technical assistance.

## Author Contributions

**Conceptualization:** Zhen Liu, Renhai Peng.

**Data curation:** Zhen Liu.

**Formal analysis:** Zhaoguo Li.

**Methodology:** Zhen Liu, Renhai Peng.

**Software:** Zhaoguo Li, Pengtao Li.

**Writing – original draft:** Zhaoguo Li.

**Writing – review & editing:** Yangyang Wei, Yuling Liu, Linxue Xing, Mengjie Liu, Pengtao Li, Quanwei Lu, Renhai Peng.

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
