## [Decision Letter · Decision Letter 0]

1 Apr 2021

PONE-D-21-08015

Genome-wide identification of the MIOX gene family and their expression profile in cotton development and response to abiotic stress

PLOS ONE

Dear Dr. Li,

Thank you for submitting your manuscript to PLOS ONE. After careful consideration, we feel that it has merit but does not fully meet PLOS ONE’s publication criteria as it currently stands. Therefore, we invite you to submit a revised version of the manuscript that addresses the points raised during the review process.

At-least following points should be addressed.

An in-depth bioinformatics analysis is suggested to gain insights in the evolution of *MIOX* genes in cotton.Proper nomenclature should be followed throughout the manuscript to describe the *MIOX* genes and MIOX proteins.The results should be clearly stated and should be discussed separately in detail.The results of transcriptome analysis should be verified with wet-lab experiments, by quantifying the expression of all the genes by Quantitative Real time PCR under various abiotic and biotic stresses.

We look forward to receiving your revised manuscript.

Kind regards,

Rana Muhammad Atif

Academic Editor

PLOS ONE

Journal Requirements:

2. Please amend the manuscript submission data (via Edit Submission) to include author Renhai Peng

Additional Editor Comments (if provided):

Reviewers' comments:

Reviewer's Responses to Questions

**Comments to the Author**

1. Is the manuscript technically sound, and do the data support the conclusions?

Reviewer #1: Partly

Reviewer #2: Yes

2. Has the statistical analysis been performed appropriately and rigorously? 

Reviewer #1: Yes

Reviewer #2: Yes

3. Have the authors made all data underlying the findings in their manuscript fully available?

Reviewer #1: Yes

Reviewer #2: Yes

4. Is the manuscript presented in an intelligible fashion and written in standard English?

Reviewer #1: Yes

Reviewer #2: Yes

5. Review Comments to the Author

Reviewer #1: This manuscript entitled “Genome-wide identification of the MIOX gene family and their expression profile in cotton development and response to abiotic stress” is a routine bioinformatics prediction of MIOX genes, their distribution in different chromosomes, their gene structure, their phylogeny, cis-regulatory elements and insilico expression in cotton species. The most important deficiency of the study is the lack of wet-lab analysis (RT-qPCR) therefore, this manuscript is incomplete. The problems and specific

comments are as follows:

1.Grammar mistakes: Line 48-50, Line 66-69, line 91-95, 2.Line 78: all

of the four diploid species contained 6 MIOX genes? Three diploid species.

3.What is Gossypioides kirkii? Why choose it? The result should be put

in the abstract section.

4.Line 82-86: The authors renamed the MIOX genes. If the renamed MIOX

gene has recognized name, how to deal with it?

5.Line 173: GhMIOX02 and GhMIOX08 should be italic.

6.Line 173-174: It is not obvious from the heat-map that “the expression

of GhMIOX02 and GhMIOX08 showed up-regulation in the early time points

(1h~3h), then down-regulation (3h~6h), and then another up-regulation

(6h~24h).” 7.Line 174-175: The expression pattern was not “While

GhMIOX03 was up-regulated in the early time points (1h~6h), but down

regulated in the late time points (12h~24h) under code stress.”

8.Line 176-178: It is not obvious from the heat-map that“Moreover,

GbMIOX02 showed continuously down-regulation at all of the time points

under salt stress.”

9.Add q-PCR verification for the transcriptome results

10.Line 194-201: This part should be present in the result section even

if there is no none of MIOX gene was amplified by transposable elements.

11.Line 211-216: There is no ka/ks analysis in the result section.

12. The method should be present in detail such as the genome version.

Reviewer #2: This manuscript entitled “Genome-wide identification of the MIOX gene family and their expression profile in cotton development and response to abiotic stress” deals with general bioinformatics prediction of MIOX genes, their distribution in different chromosomes, their gene structure, their phylogeny, cis-regulatory elements and insilico expression in cotton species. Overall, this work presented a regular meta-analysis of MIOX gene family using public available data. The overall logic of the study is relatively clear. However, this manuscript is still lacking in some points. The specific comments are as follows:

Introduction:

1: Grammar mistakes: Line 48-50, “AtMIOX4 over expression can”,

2: Line 66-69, delete “were analyzed”.

Materials and Methods:

3: Cannot see any methodology for multiple sequence alignment.

4: It is suggested to verify the results with we-lab experiments for some genes as q-PCR verification for the transcriptome results that play an important role in stress.

5: The method should be present in detail such as the genome version.

Results:

Generally, the Results are weakly described.

6: Line 78 “all of the four diploid species contained 6 MIOX genes”. Hence there are three diploid species as indicated in abstract too.

7: Further Line 78 has grammatical mistakes. Past and present tenses are in same sentence.

8: Line 91: “protein from” not “protein form”

9: Line 91-95: confusing sentence.

10: Authors should try to annotate the motifs identified through MEME to gain insights in the functions of these motifs.

Discussion:

Very week discussion of the results. Some results are not even discussed in this section.

11: Line238 has a mistake: “using a E value of 1.0E-3” etc. The full manuscript needs to be carefully reviewed and modified.

12: Line 194-204: This part should describe in the result section too even if there is no MIOX gene was amplified by transposable elements.

13: Line 211-216: There is no ka/ks analysis description in results section.

Such weaknesses in the manuscript reduce the readability for general readers and affect the scientific significance of research.

6. PLOS authors have the option to publish the peer review history of their article (what does this mean?). If published, this will include your full peer review and any attached files.

Reviewer #1: No

Reviewer #2: **Yes: **

---

## [Author Response · Author response to Decision Letter 0]

27 May 2021

Dear Editors and Reviewers:

Thank you for your letter and for the reviewers’ comments concerning our manuscript entitled “Genome-wide identification of the MIOX gene family and their expression profile in cotton development and response to abiotic stress” (Manuscript Number: PONE-D-21-08015). Those comments are all valuable and very helpful for revising and improving our paper, as well as the important guiding significance to our researches. We have studied comments carefully and have made correction which we hope meet with approval.

At present, there are only Gossypium hirsutum TM-1 materials in our laboratory, to quantify the expression of the genes by Quantitative Real time PCR, we replace Gossypium hirsutum ZM24 genome with Gossypium hirsutum TM-1 genome in this paper. In addition, we have also made a thorough revision of the graphical presentation of the data in the manuscript. Revised portion are marked in red in the “Revised Manuscript with Track Changes”. The main corrections in the paper and the responds to the reviewer’s comments are as flowing: 

At-least following points should be addressed.

1. An in-depth bioinformatics analysis is suggested to gain insights in the evolution of MIOX genes in cotton.

Response: Considering the reviewer’s suggestion, we have re-written the discussion part and the evolution of MIOX genes was supplemented.

2. Proper nomenclature should be followed throughout the manuscript to describe the MIOX genes and MIOX proteins.

Response: We have made correction according to the reviewer’s comments.

3. The results should be clearly stated and should be discussed separately in detail.

Response: We have re-written the results and discussion part according to the reviewer’s suggestion. The MIOX gene amplified by transposable elements and the ka/ks analysis were adjusted to the result part.

4. The results of transcriptome analysis should be verified with wet-lab experiments, by quantifying the expression of all the genes by Quantitative Real time PCR under various abiotic and biotic stresses.

Response: We have quantified the expression of Gossypium hirsutum MIOX genes under cold, PEG, and NaCl treatments, and added the corresponding content in the method and result part of the manuscript.

Reviewer #1:

1. Grammar mistakes: Line 48-50, Line 66-69, line 91-95. 

Response: We are very sorry for our incorrect writing, please refer to “Revised Manuscript with Track Changes” for details.

Line 48-50: The “is” changed to “was”.

Line 66-69: We adjusted the sentence structure.

line 91-95: We have revised the sentence.

2. Line 78: all of the four diploid species contained 6 MIOX genes? Three diploid species.

Response: The four diploid species are Gossypium arboretum, Gossypium herbaceum, Gossypium raimondii and Gossypioides kirkii. Since Gossypioides kirkii is not Gossypium genus, we rewrite the sentence as follows to make the expression more accurate. we also made modifications in the manuscript.

Furthermore, we found that all of the three diploid Gossypium species and the Gossypioides kirkii contained 6 MIOX genes, the number of MIOX genes in tetraploid cotton is roughly twice that of diploid cotton, but the three wild tetraploid cottons (G. tomentosum, G. mustelinum and G. darwinii) missed a MIOX gene.

3. What is Gossypioides kirkii? Why choose it? The result should be put in the abstract section.

Response: Gossypioides is a sister genus of Gossypium. In order to better illustrate the characteristics of MIOX gene family in cotton, we used the data of Gossypioides kirkii in our research, and we have put the result in the abstract section according to the Reviewer’s suggestion.

4. Line 82-86: The authors renamed the MIOX genes. If the renamed MIOX gene has recognized name, how to deal with it?

Response: The length of sequence ID in genome annotation varies greatly among different species, the short ID is only 10 characters, while the long ID is 48 characters, which caused a lot of trouble to the preparation of the paper. So we renamed the sequences. It is really true as Reviewer suggested that the renamed MIOX gene might have recognized name, to deal with this problem, “the new names of the sequence are only used in this study” was added to the end of the section “Identification of the MIOX Gene Family”.

5.Line 173: GhMIOX02 and GhMIOX08 should be italic.

Response: Considering the reviewer’s other suggestion, we have re-written this part and this sentence has been deleted.

6.Line 173-174: It is not obvious from the heat-map that “the expression of GhMIOX02 and GhMIOX08 showed up-regulation in the early time points (1h~3h), then down-regulation (3h~6h), and then another up-regulation (6h~24h).” 

Response: It is really true as reviewer suggested that heat-map cannot clearly show the up or down regulation of genes. We have re-written this part, please refer to “Revised Manuscript with Track Changes” for details. 

7.Line 174-175: The expression pattern was not “While GhMIOX03 was up-regulated in the early time points (1h~6h), but down regulated in the late time points (12h~24h) under code stress.”

Response: It is really true as reviewer suggested that heat-map cannot clearly show the up or down regulation of genes. We have re-written this part, please refer to “Revised Manuscript with Track Changes” for details. 

8.Line 176-178: It is not obvious from the heat-map that “Moreover, GbMIOX02 showed continuously down-regulation at all of the time points under salt stress.”

Response: It is really true as reviewer suggested that heat-map cannot clearly show the up or down regulation of genes. We have re-written this part, please refer to “Revised Manuscript with Track Changes” for details.

9. Add q-PCR verification for the transcriptome results.

Response: We have quantified the expression of Gossypium hirsutum MIOX genes under cold, PEG, and NaCl treatments, and added the corresponding content in the method and result part of the manuscript.

10.Line 194-201: This part should be present in the result section even if there is no none of MIOX gene was amplified by transposable elements.

Response: We have made correction according to the Reviewer’s comments.

11.Line 211-216: There is no ka/ks analysis in the result section.

Response: The ka/ks analysis was in the discussion part, and we have adjusted the ka/ks analysis to the result part and made some modifications.

12. The method should be present in detail such as the genome version.

Response: We have re-written the method part according to the reviewer’s suggestion. genome version was added, and more detailed genome version information can be referred to the corresponding references. Furthermore, “transposable elements expansion analysis of MIOX genes” and “RNA isolation and real-time PCR analysis of MIOX genes” were also added to the method part.

Reviewer #2:

1: Grammar mistakes: Line 48-50, “AtMIOX4 over expression can”,

Response: We are very sorry for our negligence of the grammar mistake, “express” were corrected as “expression”.

2: Line 66-69, delete “were analyzed”.

Response: We are very sorry for our incorrect writing, we have made correction according to the reviewer’s suggestion, please refer to “Revised Manuscript with Track Changes” for details.

Materials and Methods:

3: Cannot see any methodology for multiple sequence alignment.

Response: It is really true as reviewer suggested that there is no description for multiple sequence alignment in method part, and “The ClustalX was used to align MIOX protein sequences” was added to the part of method.

4: It is suggested to verify the results with we-lab experiments for some genes as q-PCR verification for the transcriptome results that play an important role in stress.

Response: We have quantified the expression of Gossypium hirsutum MIOX genes under cold, PEG, and NaCl treatments, and added the corresponding content in the method and result part of the manuscript.

5: The method should be present in detail such as the genome version.

Response: We have re-written the method part according to the reviewer’s suggestion. genome version was added, and more detailed genome version information can be referred to the corresponding references. Furthermore, “transposable elements expansion analysis of MIOX genes” and “RNA isolation and real-time PCR analysis of MIOX genes” were also added to the method part.

Results:

Generally, the Results are weakly described.

6: Line 78 “all of the four diploid species contained 6 MIOX genes”. Hence there are three diploid species as indicated in abstract too.

Response: The four diploid species are Gossypium arboretum, Gossypium herbaceum, Gossypium raimondii and Gossypioides kirkii. Since Gossypioides kirkii is not Gossypium genus, we rewrite the sentence as follows to make the expression more accurate. we also made modifications in the manuscript and the abstract.

Furthermore, we found that all of the three diploid Gossypium species and the Gossypioides kirkii contained 6 MIOX genes, the number of MIOX genes in tetraploid cotton is roughly twice that of diploid cotton, but the three wild tetraploid cottons (G. tomentosum, G. mustelinum and G. darwinii) missed a MIOX gene.

7: Further Line 78 has grammatical mistakes. Past and present tenses are in same sentence.

Response: We are very sorry for our incorrect writing, the “is” were corrected as “was”.

8: Line 91: “protein from” not “protein form”

Response: We are very sorry for our incorrect writing, the “form” were corrected as “from”.

9: Line 91-95: confusing sentence.

Response: We are very sorry for our incorrect writing, we have revised the sentence, please refer to “Revised Manuscript with Track Changes” for details.

10: Authors should try to annotate the motifs identified through MEME to gain insights in the functions of these motifs.

Response: The MEME suite can discovery sequence motifs, but I don't know how to annotate the motifs identified by MEME. I also think it's very important to understand the functions of these motifs, and we will try to find a suitable method, but we may not be able to complete this task in the near future.

Discussion:

Very week discussion of the results. Some results are not even discussed in this section.

11: Line238 has a mistake: “using a E value of 1.0E-3” etc. The full manuscript needs to be carefully reviewed and modified.

Response: We are very sorry for our incorrect writing, the “using a E value of 1.0E-3” was corrected as “with an E-value≤1.0E−3”. Considering the Reviewer’s suggestion, we have reviewed the full manuscript.

12: Line 194-204: This part should describe in the result section too even if there is no MIOX gene was amplified by transposable elements.

Response: We have made correction according to the Reviewer’s comments.

13: Line 211-216: There is no ka/ks analysis description in results section.

Response: The ka/ks analysis is in the discussion part, and we have adjusted the ka/ks analysis to the result part and made some modifications.

 We tried our best to improve the manuscript and made some changes in the manuscript. These changes will not influence the content and framework of the paper. And here we did not list the changes but marked in red in revised paper. We appreciate for Editors/Reviewers’ warm work earnestly, and hope that the correction will meet with approval.

Once again, thank you very much for your comments and suggestions.

Yours sincerely

---

## [Decision Letter · Decision Letter 1]

21 Jun 2021

Genome-wide identification of the MIOX gene family and their expression profile in cotton development and response to abiotic stress

PONE-D-21-08015R1

Dear Dr. Peng,

We’re pleased to inform you that your manuscript has been judged scientifically suitable for publication and will be formally accepted for publication once it meets all outstanding technical requirements.

Kind regards,

Rana Muhammad Atif

Academic Editor

PLOS ONE

Additional Editor Comments (optional):

Reviewers' comments:

Reviewer's Responses to Questions

**Comments to the Author**

1. If the authors have adequately addressed your comments raised in a previous round of review and you feel that this manuscript is now acceptable for publication, you may indicate that here to bypass the “Comments to the Author” section, enter your conflict of interest statement in the “Confidential to Editor” section, and submit your "Accept" recommendation.

Reviewer #1: All comments have been addressed

Reviewer #2: All comments have been addressed

2. Is the manuscript technically sound, and do the data support the conclusions?

Reviewer #1: Yes

Reviewer #2: Yes

3. Has the statistical analysis been performed appropriately and rigorously? 

Reviewer #1: Yes

Reviewer #2: Yes

4. Have the authors made all data underlying the findings in their manuscript fully available?

Reviewer #1: Yes

Reviewer #2: Yes

5. Is the manuscript presented in an intelligible fashion and written in standard English?

Reviewer #1: Yes

Reviewer #2: Yes

6. Review Comments to the Author

Reviewer #1: (No Response)

Reviewer #2: the Authors have addressed all of the suggestions and have improved the quality and readability of manuscript.

7. PLOS authors have the option to publish the peer review history of their article (what does this mean?). If published, this will include your full peer review and any attached files.

Reviewer #1: No

Reviewer #2: **Yes: **Dr. Muhammad Abdul Rehman Rashid

---

## [Editor Report · Acceptance letter]

28 Jun 2021

PONE-D-21-08015R1 

Genome-wide identification of the MIOX gene family and their expression profile in cotton development and response to abiotic stress 

Dear Dr. Peng:

I'm pleased to inform you that your manuscript has been deemed suitable for publication in PLOS ONE. Congratulations! Your manuscript is now with our production department. 

Kind regards, 

on behalf of

Dr. Rana Muhammad Atif 

Academic Editor

PLOS ONE